



**Taiwan Earth System Model Version 1: Description and Evaluation of Mean**
**State**
Wei-Liang Lee[1], Yi-Chi Wang[1], Chein-Jung Shiu[1], I-chun Tsai[1], Chia-Ying Tu[1], Yung-Yao Lan[1],
Jen-Ping Chen[2], Hau-Lu Pan[3], and Huang-Hsiung Hsu[1]
[1] Research Center for Environmental Changes
[2] National Taiwan University
[3] National Center for Environmental Protection
Correspondence: Wei-Liang Lee (leelupin@gate.sinica.edu.tw)



**Abstract.**
The Taiwan Earth System Model (TaiESM) version 1 is developed based on Community Earth
System Model version 1.2.2 of National Center for Atmospheric Research. Several innovated
physical and chemical parameterizations, including trigger functions for deep convection, cloud
macrophysics, aerosol, and three-dimensional radiation–topography interaction, as well as a one-
dimensional mixed-layer model optional for the atmosphere component, are incorporated. The
precipitation variability, such as diurnal cycle and propagation of convection systems, is improved in
TaiESM. TaiESM demonstrates good model stability in the 500-year preindustrial simulation in
terms of the net flux at the top of the model, surface temperatures, and sea ice concentration. In the
historical simulation, although the warming before 1935 is weak, TaiESM well captures the
increasing trend of temperature after 1950. The current climatology of TaiESM during 1979–2005 is
evaluated by observational and reanalysis datasets. Cloud amounts are too large in TaiESM, but their
cloud forcing is only slightly weaker than observational data. The mean bias of the sea surface
temperature is almost zero, whereas the surface air temperatures over land and sea ice regions exhibit
cold biases. The overall performance of TaiESM is above average among models in Coupled Model
Intercomparison Project phase 5, particularly that the bias of precipitation is smallest. However,
several common discrepancies shared by most models still exist, such as the double Intertropical
Convergence Zone bias in precipitation and warm bias over the Southern Ocean.



## 1. Introduction

The Earth system model (ESM) is a state-of-the-art tool that can simulate the long-term
evolution of the climate system including the atmosphere, ocean, land, and cryosphere and provide
future projections from the scientific aspect to study the impact of global climate change on the
natural environment, ecosystem, and human society (IPCC 5th Assessment Report, 2013). Because
of the constraint of computing power, the spatial resolution of ESMs participated in the Coupled
Model Intercomparison Project Phase 5 (CMIP5; Taylor et al. 2012) is generally on the order of
approximately 100 km. However, this coarse resolution is unsuitable for climate studies in the
Taiwan area because this island is 400 km long and 150 km wide, which occupies only several grid
boxes in these ESMs. For the Taiwanese scientific community, developing a global model to provide
climate data in various future scenarios with high temporal resolutions—daily or hourly—for
dynamical or statistical downscaling is desirable. Taiwan's National Science Council (now Ministry
of Science and Technology) has accordingly launched a project to increase climate modeling
capability and capacity in Taiwan, the core component of which is Taiwan Earth System Model
(TaiESM) development.
In Taiwan, manpower and expertise for climate research are limited; thus, we could not create an
ESM from scratch. Therefore, TaiESM version 1 is developed on the basis of the Community Earth
System Model version 1.2.2 (CESM1.2.2; Hurrell et al., 2013) from National Center for
Atmospheric Research (NCAR) sponsored by National Science Foundation and the Department of
Energy of the United States. TaiESM consists of the Community Atmosphere Model version 5.3
(CAM5), Community Land Model version 4 (CLM4), Parallel Ocean Program version 2 (POP2), and
Community Ice Code version 4 (CICE4). We replace or modify existing parameterizations in CAM5,
including new trigger functions for the deep convection scheme (Wang et al., 2015), new cloud
macrophysics scheme for cloud fraction calculation (Wang et al., 2018, Shiu et al., 2018), and a
three-moment aerosol scheme (Chen et al., 2013). A novel parameterization for the impact of three-





dimensional (3D) radiation–topography interactions (Lee et al., 2013) is added to CLM4. In addition,
a one-dimensional (1D) mixed-layer ocean model with a high vertical resolution (Tsuang et al., 2009)
is used for CAM5 with slab ocean simulation in TaiESM.

An object of TaiESM development is to improve the simulations of climate variability in various

spatial and temporal scales for more reliable climate projections in Taiwan. Weather and climate in
Taiwan is deeply affected by capricious East Asia/western North Pacific monsoon and typhoons. In
addition, because of its small size and steep terrain, predicting the frequencies of severe weather and
heavy precipitation in Taiwan is highly difficult (Hsu et al., 2011). Therefore, the parameterizations
selected for TaiESM are for enhancing variability simulation. The trigger functions for the deep
convection scheme in TaiESM, adopted from National Centers for Environmental Prediction (NCEP)
Global Forecast System (GFS) with Simplified Arakawa–Schubert scheme (SAS; Pan and Wu, 1995;
Han and Pan, 2011), aim to improve the timing of convective precipitation occurrence. As
demonstrated by Lee et al. (2008), by using GFS, these trigger functions are key to improved
simulations of the diurnal rainfall cycle over the Southern Great Plains (SGP) in the United States.
The parameterization for 3D radiation-topography interactions account for the effects of shadows
and reflections from subgrid topographic variation on the surface solar flux (Lee et al., 2011) for
application to general circulation models (GCMs). The high-resolution 1D mixed-layer model can
resolve fast change in the skin temperature of the sea surface (Tu et al., 2005).

The organization of this paper is as follows: Section 2 describes TaiESM, particularly the new

and modified schemes different from CESM1.2.2. Section 3 presents the design of model
experiments. Sections 4 and 5 provide the description of TaiESM performance in preindustrial and
historical simulations, respectively. Summary and conclusions are given in Section 6.

**2. Model description**

The development of TaiESM is based on CESM1.2.2, in which the ocean, sea ice, and river





components, as well as the infrastructure of the model, remain unchanged. For the atmosphere,
several physical and chemical parameterizations are modified, as two trigger functions are added to
the default deep convection scheme, and cloud macrophysics and aerosol schemes are replaced. A
parameterization of surface albedo adjustment is added to CLM4 to account for the topographic
effect on surface solar radiation. In addition, a 1D mixed-layer ocean model is integrated to TaiESM
for simulations of CAM5 coupled with a slab ocean.

**2.1. Atmosphere**
The atmosphere model in TaiESM is based on CAM version 5.3 (Neale et al., 2010). The
dynamic core is finite volume (Lin, 2004) in a hybrid sigma-pressure vertical coordinate. The Rapid
Radiative Transfer Model for GCMs (RRTMG; Iacono et al., 2008) with two-stream approximation,
correlated $k$-distribution, and Monte Carlo Independence Column Approximation (McICA; Pincus et
al., 2003) is employed to calculate radiative fluxes and heating rates in the atmosphere. The shallow
convection and moist turbulence schemes are based on those reported by Park and Bretherton (2009)
and Bretherton and Park (2009), respectively. A two-moment cloud microphysics scheme (Morrison
and Gettelmen, 2008) is used to predict changes in the mass and number of cloud droplets and to
diagnose stratiform precipitation.

**2.1.1. Trigger function for deep convection**
Convective trigger function is a critical part of the cumulus parameterization scheme to
determine the initiation of precipitating convection and thus has a critical role in rainfall variability
simulation. With the Zhang–McFarlane scheme framework (Zhang and McFarlane, 1999; Neale et
al., 2008), TaiESM has adopted two convection triggers proposed by Wang et al. (2015): unrestricted
launching level (ULL) and convective inhibition (CIN). Wang et al. (2015) reported significant
improvements in the diurnal rainfall peak at the Atmospheric Radiation Measurement (ARM) SGP



site, mainly because of the suppression of daytime spurious convection by the CIN trigger and
initiation of nighttime mid-level convection by ULL trigger. ULL may also aid in improving diurnal
rainfall phase in many other areas worldwide when implemented in the newly developed Energy
Exascale Earth System Model version 1 (E3SMv1) of the U.S. Department of Energy (Xie et al.,

2019).

Similar to that in GFS, improvement in the diurnal rainfall cycle is found in TaiESM. Figure 1

displays local times (LTs) of the diurnal rainfall peak occurrence, referred to as the peak phase from
the 11-year (2001–2011) Tropical Rainfall Measuring Mission (TRMM) merged satellite data
(Huffman et al. 2007) and the historical model runs during 1979–2005. Two distinct changes in
diurnal rainfall cycle are found in TaiESM compared with those in CESM1.2.2. First, the diurnal
rainfall peak over the tropical lands, such as the Central Africa and the Amazon basin, are delayed to
14–18 LT from the 12–14 LT peak phase of CESM1.2.2. A similar delay is also observed in islands
such as Borneo. Second, nocturnal rainfall in TaiESM is increased compared with that in CESM1.2.2,
particularly in coastal and topographical regions where propagating convective organizations
emitting from the coastline or topographical regions (Kikuchi and Wang, 2010), demonstrated as the
gradual phase change in Figure 1, such as the eastern slope of the Rocky Mountains.

Figure 2 shows the Hovmöller diagram of longitude and local time for TaiESM, CESM1.2.2,

and TRMM observations over SGP (35°N–40°N, 90°W–110°W). Convection occurs at 104°W in the
evening and propagates eastward in the observation (Carbone and Tuttle, 2008). In CESM1.2.2,
convection occurs in the early afternoon and peaks before midnight, but it is stationary at the same
location. TaiESM successfully captures the eastward propagation of the rainfall and a better
occurrence time of convection in the late afternoon, as well as the more realistic rainfall intensity.
This result is consistent with the single-column model tests of Wang et al. (2015), indicating that
their proposed convective trigger may be the cause of these improvements. Furthermore, Wang and
Hsu (2019) demonstrate that the improvement of nocturnal rainfall over SGP is mainly from the



superior response of the ULL + CIN convective trigger to the low-level convergence between the
branch of mountain-plain solesoid and low-level jet from Gulf of Mexico. With the horizontal
resolution at an order of 100 km, this result suggests that the convective trigger of TaiESM captures
the large-scale preconditioning associated with the convective organization there (Dirmeyer et al.,
2011), rather than only the convective systems itself.

**2.1.2. Cloud fraction**

The cloud macrophysics scheme used in TaiESM is the GFS–TaiESM–Sundqvist (GTS) scheme.

It was first developed for the NCEP GFS model and has been further used for the TaiESM. Similar
to that in many numerical weather prediction and global climate models, the GTS scheme is based on
the Sundqvist scheme (Sundqvist et al., 1989), which calculates changes in cloud condensates in a
grid box on the basis of the budget equation for relative humidity (RH) with large-scale advection.
The CAM5 macrophysics (Park et al., 2014) follows this approach and assumes empirical values of
critical RH ($RH_c$) as the threshold of condensation. The key difference of the GTS scheme from the
CAM5 macrophysics is the re-derivation of the equation relating the change in the subgrid-scale
cloud condensate using the distribution width of mixing ration of total water ($q_t$) to replace $RH_c$, as
indicated in Tompkins (2005). The unnecessary use of $RH_c$ is consequently removed to allow an
improved correlation among cloud fraction, RH, and condensates.

Figure 3 illustrates cloud fraction as a function of RH of water vapor ($q_v/q_s$) and RH of

condensates ($q_l/q_s$) for the CAM5 macrophysics and the GTS schemes with uniform and triangular
probability density functions (PDFs) of $q_t$ in a grid box. Given the same RH of water vapor, the PDF-
based calculation allows larger cloud fraction if more cloud condensates exist in the grid than the
CAM5 macrophysics. The difference in cloud fraction produced by two PDFs is small, implying that
this scheme might not be very sensitive to the shape of the distribution. The triangular PDF provide
additionally rapid changes in cloud fraction when the RH of condensates and water vapor changes,



and it is used as the default PDF of the GTS scheme.

### 2.1.3. Aerosol

The aerosol parameterization used in TaiESM is the Statistical-Numerical Aerosol
Parameterization (SNAP; Chen et al., 2013). SNAP is a bulk parameterization, and the modal
approach (Seigneur et al., 1986; Whitby and McMurry, 1997) is adapted to describe the particle size
distribution. In contrast to conventional aerosol parameterizations in most ESMs, changes in the
zeroth moment (number), second moment (surface area), and third moment (mass) due to physical
processes are tracked in SNAP. The physical processes included in SNAP are emission, nucleation,
coagulation, condensation, mixing, as well as dry and wet deposition. SNAP has been applied to the
US EPA Models-3/Community Multi-scale Air Quality (CMAQ; Byun and Schere, 2006) modeling
system and been verified by observations (Chen et al., 2013; Tsai et al., 2015) with Weather
Research and Forecasting Model (WRF; Skamarock et al., 2008).

### 2.2. Land

The land model in TaiESM is CLM4 (Oleson et al., 2010; Lawrence et al., 2011). The surface
albedo is primarily a function of vegetation, soil moisture, solar zenith angle, as well as snow
reflectivity calculated by the Snow, Ice, and Aerosol Radiative Model (SNICAR; Flanner and Zender,
2006), which considers the aerosol deposition of black carbon and dust, effective size of snow grains,
and vertical profile of heating. As the albedo of a grid box is determined, it is then adjusted to
include the topographic effect on surface solar radiation.
The parameterization for 3D radiation–topography interactions is to evaluate the impact of
topography on surface solar radiation, including insolation on various slopes and aspects, shadow
cast by nearby mountains, and reflections between surfaces (Lee et al., 2013). It is developed on the
basis of the numbers of "exact" Monte Carlo calculation that simulates the scattering, reflection, and

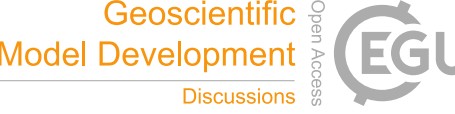

absorption of photons within the 3D atmosphere and surface (Chen et al., 2006; Liou et al., 2007;
Lee et al., 2011). The parameterization adjusts surface albedo so that the solar radiation absorbed by
the surface in the land model corresponds with the results of the Monte Carlo calculation. Several
topographic variables are used for input, including the slope, aspect, sky view factor, terrain
configuration factor, standard deviation of elevation within a grid box, and solar zenith and azimuth
angles. Gu et al. (2012) and Liou et al. (2013) demonstrate that this topographic effect can increase
the amount of snowpack in the valley and enhance the snowmelt in mountains in the WRF
simulations over the western United States. Lee et al. (2015, 2019) also demonstrate that
incorporating this parameterization to the Community Climate System Model version 4 (CCSM4)
can significantly improve the surface energy budget over the Rocky Mountains and the Tibetan
Plateau and thus reduce the systematic cold bias in the CMIP5 models.

**2.3. Ocean and sea ice**

The sea ice and dynamic ocean components of TaiESM are from the CICE4 (Hunke and

Lipscomb, 2008) and POP2 (Smith et al., 2010) of Los Alamos National Laboratory, respectively.
The CICE4 and POP2 configurations in the fully coupled TaiESM simulations are identical to those
in CESM1.2.2. To save computational resources, a zero-dimensional slab ocean model without
dynamical process is commonly used to simulate the thermodynamic interaction between the
atmosphere and ocean. In TaiESM, a 1D mixed-layer model is coupled with the atmosphere
component to reveal the impact of the fast evolution in upper ocean layers.

The one-column ocean model Snow–Ice–Thermocline (SIT; Tu and Tsuang, 2005; Tsuang et al.

2009) is designed to simulate the sea surface temperature (SST) and upper ocean temperature
variations with a high vertical resolution, including cool skin, diurnal warm layer, and mixed-layer of
the upper ocean. SIT calculates changes in temperature, momentum, salinity, and turbulent kinetic
energy driven by vertical fluxes parameterized using the classical K approach. Cool skin is derived





by considering merely molecular transport for vertical diffusion of heat in the skin layer, where the
skin layer thickness is calculated as described by Artale et al. (2002). Beneath the skin layer, eddy
diffusivity is determined according to a second-order turbulence closure approach (Gaspar et al.,
1990), and the 1-m vertical discretization is deployed down to a 10-m depth for resolving diurnal
warm layer. Because of the lack of ocean circulation in the one-column ocean model, the calculated
ocean temperatures are weakly nudged to climatology for ocean below 10-m depth to avoid climate
drift. SIT and AGCM exchange SST and fluxes at every time step in tropics (30°S–30°N), whereas
climatological SST drives the AGCM elsewhere. Note that SIT is not integrated with the dynamic
ocean model (POP2); therefore, fully coupled TaiESM simulations do not include SIT.

**3. Experiment design**

The horizontal resolution of the atmosphere and land in TaiESM is 0.9° latitude by 1.25°

longitude, with 30 vertical layers in the atmosphere. The ocean and sea ice components use the same
horizontal resolution with $320 \times 384$ grid points (approximately 1°) and 60 vertical layers in the
ocean. Currently, TaiESM is calibrated only to this set of resolutions, in which several microphysical
properties of clouds are modified to minimize radiation imbalance at the top of the atmosphere
(TOA). Additional model tuning would be required for stable simulations at higher or lower
resolutions.

TaiESM is spun-up using CMIP5 preindustrial conditions, such as greenhouse gas

concentrations, surface aerosol emissions, solar constant, and land-use types. Because TaiESM is
considerably similar to CESM1.2.2, we use the model restart files of CESM1.2.2 for the 1850 control
run as the initial condition to reduce the computation effort, particularly for the ocean component
that may need more than a thousand years to reach a steady state. The spin-up integration continues
for 500 years, and the climate state at the end of year 500 is used as the initial condition for the 500-
year preindustrial control (hereafter piControl) simulation. The historical simulation then starts at the

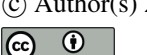

end of piControl (i.e., year 1000) with observationally based forcing, including changes in the solar
constant, greenhouse gas concentrations, surface aerosol emission, and volcanic eruptions, from
1850 to 2005.

**4. Model stability in piControl run**

In this section, the global means of several climatological variables in piControl run of TaiESM

are evaluated. The climate drift from CESM1.2.2 initial conditions to TaiESM equilibrium during the
spin-up is also assessed to represent differences between the two models caused by the new or
modified physical processes in TaiESM.

**4.1. Time series of climate states**

Figure 4 illustrates the time series of several global mean variables in TaiESM piControl. The

long-term global mean TOA net flux is 0.086 W m$^{-2}$, and it decreases by 0.0054 W m$^{-2}$ in 500 years
but insignificantly. Furthermore, the mean surface net flux is 0.081 W m$^{-2}$ with an almost identical
decreasing trend as TOA net flux. The imbalance at TOA causes heating of the whole model system,
and the less imbalance at the surface indicates a smaller part of excessive energy remains in the
atmosphere in piControl. Consequently, the long-term trend of surface air temperature (SAT) is
0.0088 K century$^{-1}$ in 500 years, which is significant. By contrast, the trend of SST is 0.0047 K
century$^{-1}$, only about half of the SAT trend and insignificant. By breaking down the surface net flux,
we found that the energy exchange between the atmosphere and land is less than $10^{-5}$ W m$^{-2}$,
whereas the net flux into the ocean is 0.114 W m$^{-2}$ (figures not shown). The excessive energy enters
the deep ocean and leads to a steady increase in global mean ocean temperature of 0.030 K century$^{-1}$.
Therefore, even after a 1000 years' simulation, the system does not reach the thermodynamic
equilibrium. In addition, considering that the heat capacity of the entire ocean is approximately 1000
times larger than the atmosphere, the heating rates of the atmosphere caused by the residual net flux





(0.005 W m$^{-2}$) is too small compared with the heating rate of the ocean. It implies that an unknown
energy leak may exist in the coupling between the atmosphere and ocean, which requires further
investigation in programming to fix this problem.

The annual mean time series of sea ice area in the Northern Hemisphere (NH) and Southern

Hemisphere (SH) are exhibited in the bottom panels of Figure 4. The Arctic sea ice has a small but
significant trend of $-0.01 \times 10^6$ km$^2$ century$^{-1}$, corresponding to the slight warming of the entire
model fairly well. By contrast, the linear trend of the sea ice area in the Southern Ocean over the
500-year span is almost zero, even though the variation is much larger. The minimal change in the
sea ice area indicates that the energy gain of the cryosphere could be negligible compared with other
model components.

The global mean sea surface salinity (SSS) reduces significantly by $-0.0036$ g kg$^{-1}$ century$^{-1}$.

However, it can be found that SSS is almost constant with a slope of about $10^{-4}$ g kg$^{-1}$ century$^{-1}$ after
year 700. On the other hand, there is a small but significant decreasing trend of the global mean
ocean salinity of $1.3 \times 10^{-4}$ g kg$^{-1}$ century$^{-1}$, which is very close to the trend of SSS in the last 300
years. This reduction is probably related to the additional freshwater flux from the decrease in Arctic
sea ice area. In addition, the long-term mean of evaporation minus precipitation (E − P) is $-1.16$ mm
day$^{-1}$, and it may also contribute to the freshening of the ocean.

**4.2. Comparison with CESM**

The long-term means of several variables in piControl runs performed by CESM1.2.2 and

TaiESM are listed in Table 1. The TOA net flux in TaiESM and CESM1.2.2 are both within 0.09 W
m$^{-2}$. The magnitude of imbalance is acceptable, but it could lead to warming of the entire Earth
system. The SAT and SST in TaiESM are higher than those in CESM1.2.2 by 0.42 and 0.23 K,
respectively. Shortwave (SW) net flux at TOA in TaiESM is larger than CESM1.2.2 by 2.24 W m$^{-2}$,
which might be the primary cause of higher surface temperatures and consequently result in larger





longwave (LW) net flux at TOA of 2.23 W m$^{-2}$. The difference in the clear-sky net SW flux at TOA
is only 0.66 W m$^{-2}$, suggesting that the surface albedo difference is small, whereas the contribution
from the difference in cloud reflection is larger. Although the high and low cloud covers in TaiESM
are larger than those in CESM1.2.2, the magnitude of SW cloud forcing (SWCF) is smaller in
TaiESM. It indicates that clouds in TaiESM are less reflective than that those in CESM1.2.2. By
contrast, the differences in clear-sky net LW flux at TOA and LW cloud forcing (LWCF) are 1.67
and 0.59 W m$^{-2}$, respectively; therefore, the warmer surface and atmosphere have greater
contribution to addition outgoing longwave radiation (OLR) in TaiESM. However, the amount of
high cloud in TaiESM is substantially larger than that in CESM1.2.2. This implies that the high
clouds in TaiESM could be optically thinner. The relation between cloud forcing and cloud cover in
SW and LW in TaiESM must be due to the GTS scheme, which can produce larger fraction but less
dense clouds compared with the cloud macrophysics scheme in CAM5.

**5. Historical simulation**
In this section, we evaluate the performance of TaiESM historical simulation with the
observation or reanalysis data. The temporal evolution of global mean temperature from the
preindustrial to present day is assessed. The mean states of the current climate, defined as the period
of 1979–2005, in the historical simulation are used for comparison.

**5.1. Global mean temperature evolution**
Figure 5 illustrates changes in global mean near-surface temperature anomaly of TaiESM and
two observations, Berkeley Earth Surface Temperature (BEST; Rohde et al., 2013) and Goddard
Institute for Space Studies Surface Temperature (GISTEMP; Lenssen et al., 2019), by using the
mean temperature of 1951–1980 as the benchmark. The warming trend of TaiESM is weaker than
the observation data during 1850–1935. The evolution of SAT in TaiESM exhibits fluctuation



similar to observations, particularly before 1900, but with smaller amplitudes. The magnitudes of
cooling induced by major volcanic eruptions, such as Krakatoa (1883), Santa Maria (1902), Agung
(1963), and Pinatubo (1991), in TaiESM is close to those in the observational data, implying that the
radiative forcing due to stratospheric aerosols is in good agreement with the observations. After 1950,
the change in SAT of TaiESM follows the observations and captures the trend of global warming
very well. The warming rate of TaiESM during 1950–2005 is 1.12 K century$^{-1}$, comparable with
1.16 and 1.27 K century$^{-1}$ of BEST and GISTEMP, respectively.

### 317 5.2. Cloud and radiation

Figure 6a demonstrates the comparison in the total cloud fraction between TaiESM and
Moderate Resolution Imaging Spectroradiometer (MODIS) Level 3 product during 2001–2012.
TaiESM overestimates the total cloud fraction by approximately 3% globally with a root mean
square difference (RMSD) of 14.07. Almost all of the Arctic Ocean is overcast in TaiESM, which is
approximately 30% higher than observational data. Cloud fraction is also severely overestimated
over the Antarctic continent and the Southern Ocean. TaiESM produces too much cloud over the
southern branch of the Intertropical Convergence Zone (ITCZ) in the central and eastern Pacific,
implying the prevalence of double ITCZ, which will be discussed in a subsequent section. Excessive
amount of clouds is also noted in the maritime continent, western equatorial Indian Ocean, and most
of the land areas. By contrast, cloud fraction is remarkably underestimated in the Amazon basin and
the subtropical ocean, particularly the stratocumulus near the western coasts of continents. Compared
with the synergic CloudSat and Cloud-Aerosol Lidar with Orthogonal Polarization (CALIOP) data
during 2006–2010 (Kay and Gettelman, 2009), low clouds in TaiESM are systematically
underestimated over the entire tropical and subtropical regions, as shown in Figure 6b, whereas they
are overestimated in high-latitude areas. The total cloud fraction in the tropics is high because of
excessive high cloud in the model (Figure 6c).



Clouds can substantially modulate the radiation field because of its high reflectivity in SW and
high absorptivity in LW. Figure 7a illustrates the comparison of SWCF in TaiESM with that in
Clouds and the Earth's Radiant Energy System–Energy Balanced and Filled data (CERES–EBAF;
Kato et al., 2018) over 2000–2015. In terms of the global mean, SWCF in TaiESM is very close to
that of the observational data by 0.19 W m$^{-2}$ larger. Although there is excessive cloud over the polar
regions, such as the Southern Ocean near the Antarctic continent and almost all of the Arctic Ocean,
in TaiESM, SWCF is not as strong as that in the observational data. It indicates that polar cloud in
TaiESM is too thin optically, probably because of the GTS cloud macrophysics scheme. In the
subtropical and tropical regions, SWCF generally follows the spatial pattern of total cloud fraction
that a larger cloud fraction produces stronger SWCF, such as the storm track in the North Pacific,
southern branch of ITCZ, maritime continent, western tropical Indian Ocean, and south of the Sahara
Desert. However, SWCF is too strong over the Amazon basin in TaiESM, even though there is
underestimated amount of clouds. By contrast, because of underestimated total cloud fraction, SWCF
in TaiESM is too weak over the stratocumulus areas off the California and Peru coasts as well as
over the subtropical Pacific, Atlantic, and Indian Oceans in the SH.
The global mean of LWCF in TaiESM is significantly weaker than that in CERES–EBAF by
4.31 W m$^{-2}$. As illustrated in Figure 7b, TaiESM underestimates LWCF worldwide, and the
magnitude of LWCF bias generally follows the bias of high cloud. Positive LWCF bias only exists in
some regions over the tropical ocean with too many high clouds in TaiESM. However, although
more high clouds exist along the northern branch of ITCZ, LWCF is weaker in the model. The
remarkable negative LWCF bias seems incompatible with the overestimated high clouds because
more high clouds should be able to intercept more LW radiation from the surface. This inconsistency
is probably due to the lower altitude of the high clouds or the less dense clouds in TaiESM.

**5.3. Surface temperature**





Figure 8a illustrates the comparison of SST between TaiESM and Hadley Centre Sea Ice and
Sea Surface Temperature dataset (HadISST; Rayner et al., 2003). The regions with a long-term mean
sea ice concentration larger than 15% are not used for calculations of the mean and RMSD. The
global mean bias of SST in TaiESM is 0.01 K with an RMSD of 1.05 K. The overestimated SST
over the Southern Ocean and subtropical South Pacific is probably induced by additional downward
SW radiation because of the inaccurate microphysical properties of polar clouds (Kay et al., 2016)
and the negative bias of cloud fraction as shown in Figure 6a. The warm bias in the major upwelling
regions off the western coasts of Americas and Africa is a common deficiency in many climate
models (Griffies et al., 2009), caused by insufficient spatial resolution of the atmosphere and ocean.
Warm bias can also be found in North Atlantic including the coast of North America, Labrador Sea,
and south of Greenland. Negative biases exist in most of the North Pacific and subtropical North
Atlantic, probably because of overestimated wind stress in these regions.
Although the SST bias in TaiESM is very small, the global mean SAT in TaiESM is
substantially colder than the observational data by 0.49 K with an RMSD of 1.68 K. This result
indicates that the temperature over land and sea ice in TaiESM is severely underestimated (Figure
8b). Cold bias exists over most of the polar regions, the Tibetan Plateau, and tropical land areas (e.g.,
Amazonia, Central Africa, and Southeast Asia). It must be due to the excessive cloud that reflects
excessive sunlight. SAT bias over the ocean generally follows SST bias, except that the SAT bias in
the subtropical South Pacific is very small despite the warm SST bias.

**5.4. Precipitation**
Figure 9 illustrates the mean precipitation over 1979–2005 in TaiESM and Global Precipitation
Climatology Project (GPCP; Huffman et al., 2009) 1-Degree Daily (1-DD) data. TaiESM
overestimates the global precipitation by 0.38 mm day$^{-1}$ with an RMSD of 1.11 mm day$^{-1}$. The most
pronounced bias in TaiESM is the double ITCZ—a common issue in most contemporary GCMs (Lin,



2007, Hirota and Takayabu, 2013) and in CESM1.2.2 (Wang et al., 2015). The precipitation rates of
both the northern and southern ITCZ branches are extremely strong. The overly intense convection
strengthens the subsidence and consequently produces too little rainfall along the equator.
Precipitation is also overestimated in the maritime continent, while it is severely underestimated in
Borneo. In TaiESM, the land–sea contrast in precipitation is not as apparent as in the observation
over the warm pool region. The South Pacific convergence zone (SPCZ) is also too strong and too
parallel to the ITCZ. The dipole bias in the tropical Indian Ocean, excessive rainfall in the western
part and scant rainfall in the eastern part, still exists as in NCAR models (Gent et al., 2011). There is
also a double ITCZ bias in the Atlantic Ocean that the southern branch is too strong and the northern
branch is too weak. In South America, precipitation over the Amazon basin is considerably
underestimated, whereas excessive orographic precipitation can be found along the Andes (Cook et
al., 2012).

**5.5. Sea ice**
Figure 10 presents the annual mean of sea ice concentration in the Arctic Ocean and Southern
Ocean in TaiESM, and the black lines indicate the 15% mean concentration from the National Snow
and Ice Data Center (NSIDC) Climate Data Record (CDR) of passive microwave sea ice
concentration version 3 (Peng et al., 2013), during 1979–2005. In the NH, TaiESM severely
overestimates sea ice concentration over the North Pacific, particularly in the Sea of Okhotsk.
TaiESM also overestimates sea ice in the Barents Sea and near the east coast of Greenland but
slightly underestimates sea ice in Labrador Sea. In the SH, sea ice in TaiESM is generally in
agreement with the observation. Excessive sea ice is noted in the area south of New Zealand, but in
the Indian Ocean region, sea ice is scant. This deviation follows the SST bias presented previously.
Figure 11 illustrates the temporal evolution of the annual sea ice concentration in TaiESM
compared with that in the CDR. The change in NH sea ice in TaiESM generally captures the trend in



the observation before 2002. However, there is an increase in TaiESM in the last 4 years, in contrast
to an accelerated reduction in observational data. This sea ice increase could be a fluctuation in a
climate simulation, and it requires longer integration for additional investigation. In SH, a decreasing
trend of the sea ice concentration can be found in TaiESM, whereas it remains almost unchanged in
observational data. Because there is no land–sea model in TaiESM, the discharge of the ice sheet
from Antarctic continent to Southern Ocean, the major source of SH sea ice, cannot be simulated
accurately. Consequently, the sea ice concentration in the SH could be controlled primarily by
temperature in TaiESM, leading to an unrealistic temporal evolution.

**5.6. Comparison with CMIP5 models**

The overall performance of TaiESM historical simulation during 1979-2005 is evaluated by

comparing with other CMIP5 models following the metrics introduced by Gleckler et al. (2008).
Figure 12 shows the normalized space-time root-mean-square-error (RMSE) of selected variables
from TaiESM, several CMIP5 models, and multi-model ensemble (MME) against reanalysis and
observation datasets. The reference data of air temperatures (TA), zonal and meridional wind
velocities (UA and VA), and geopotential height (ZG) at various pressure levels, as well as the
surface air temperature (TAS), are from Collaborative Reanalysis Technical Environment (CREATE)
Multi-Reanalysis Ensemble version 2 (MRE2; Potter et al., 2018). The observational precipitation
(PR) data is from GPCP. Upward longwave radiation in the total sky (RLUT) and clear sky
(RLUTCS) and upward shortwave radiation in the total sky (RSUT) and clear sky (RSUTCS) are
from CERES-EBAF. It is expected that the errors of CMIP5 MME are generally the smallest.
TaiESM has smallest bias in PR among all CMIP5 models, and its performance in RSUT and RLUT
is also very good. The relative poor performance in TAS is primarily due to the cold bias over land
and sea ice areas. The RMSEs of all variables in TaiESM are smaller than the median CMIP5 error,
indicating that the performance of TaiESM is above average among all CMIP5 models. In particular,



RMSEs of PR, RLST, and RLUT of TaiESM are among the smallest

**6. Summary and conclusions**
This paper documents the TaiESM version 1, developed on the basis of CESM1.2.2, with
revised physical and chemical parameterizations, including 1) trigger functions for deep convection,
which can improve the variability simulation in convective rainfall; 2) GTS cloud macrophysics
scheme to avoid artificial RH threshold for cloud formation; 3) three-moment SNAP aerosol scheme;
4) 3D radiation–topography interactions to account for the impact of shading and reflection on
shortwave radiation in mountains. A 1D mixed-layer ocean model is incorporated to the atmosphere
component to simulate the thermodynamic air-sea interaction, but it is not used for fully coupled
simulations.
TaiESM stability is assessed using 500-year piControl. Although constant imbalance in the net
flux at the TOA exists, the drifts of global mean SAT and SST are very small, with long-term trends
of 0.0088 and 0.0047 K century$^{-1}$, respectively. The excessive energy enters the deep ocean and
leads to continuous warming by 0.030 K century$^{-1}$. The drifts in the sea ice concentration in both NH
and SH are both small because of the nearly zero net energy flux from the atmosphere to sea ice.
However, the global mean SSS and total ocean salinity both demonstrate significantly decreasing
trends.
For the historical evolution of SAT, the warming of TaiESM from 1850 to 1935 is too weak
compared with the observation. After 1950, TaiESM satisfactorily captures the trend of global
warming with a heating rate of 1.12 K century$^{-1}$ comparable to the observation of 1.16 K century$^{-1}$.
The current climatology of TaiESM during 1979–2005 is generally in agreement with the
observations. The overall performance of TaiESM is better than the median of CMIP5 models,
particularly that the RMSE of precipitation is smallest. There are too many clouds in TaiESM,
whereas the SWCF and LWCF are almost similar to and weaker than the observation, respectively.





This result implies that the new cloud macrophysics scheme produces larger amount but optically
thinner clouds. SST in TaiESM is very close to the observation, whereas SAT is significantly colder,
implying remarkably underestimated SAT over land and sea ice surfaces. TaiESM produces
excessive precipitation, and the biases of double ITCZ and dipole in the tropical Indian Ocean exist,
whereas there is a severe dry bias in the Amazon basin. The trend of the NH sea ice concentration in
TaiESM follows the observation well, whereas it might not capture the accelerating reduction in the
21st century.
This paper focuses on the evaluation of long-term climatological state and evolution of global
mean quantities in TaiESM in preindustrial and historical simulations. The other part of the
characteristics of an ESM, climate variability, is also very critical to the performance of a model, and
it requires additional in-depth research. Further investigation of climate variability in TaiESM,
including the El-Niño and Southern Oscillation, intraseasonal oscillation, monsoon, and extreme
precipitation, will be documented in the follow-up papers.

*Code and data availability.* The model code of TaiESM version 1 is available at
https://doi.org/10.5281/zenodo.3626654. Output data of TaiESM using CMIP5 forcing, including
preindustrial  and  historical  simulations,  are  available  at
http://cclics.rcec.sinica.edu.tw/index.php/databases/data.html.

*Author contributions.* HHH is the initiator and the primary investigator of the TaiESM project. WLL
is the main model developer and writes the majority part of the paper. YCW is the developer and
writer of trigger functions for deep convection. YCW and CJS are the developer and writers of cloud
macrophysics. ICT and JPC are the developers and writers of SNAP aerosol scheme. CYT and YYL
are developers of 1D mixed-layer model and CYT is the writer of this section. YLP helps develop
the theoretical basis of trigger functions for deep convection and cloud macrophysics.




*Competing interests*. The authors declare that they have no conflict of interest.

*Acknowledgements.* The contribution from WLL, YCW, CJS, ICT, CYT, YYL, and HHH to this
study is supported by Ministry of Science and Technology of Taiwan under contracts MOST 106-
2111-M-001-002, MOST 106-2111-M-034-002, and MOST 106-2111-M-001-005. JPC is also
supported by MOST 107-2111-M-001-012. We thank the computational support from National
Center for High-performance Computing of Taiwan. This manuscript is edited by Wallace Academic
Editing.

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





| Variable | CESM1.2.2 | TaiESM |
|---|---|---|
| SAT (°C) | 13.16 | 13.58 |
| SST (°C) | 19.52 | 19.75 |
| TOA net flux (W m$^{-2}$) | 0.080 | 0.089 |
| TOA net SW flux (W m$^{-2}$) | 237.79 | 240.03 |
| TOA net LW flux (W m$^{-2}$) | 237.71 | 239.94 |
| TOA clear-sky net SW flux (W m$^{-2}$) | 285.41 | 286.07 |
| TOA clear-sky net LW flux (W m$^{-2}$) | 260.35 | 262.02 |
| SWCF (W m$^{-2}$) | −47.62 | −46.05 |
| LWCF (W m$^{-2}$) | 22.67 | 22.08 |
| High cloud cover (%) | 37.81 | 45.61 |
| Low cloud cover (%) | 41.96 | 41.99 |


**Table 1.** Long-term global means of selected climatological variables from CESM1.2.2 and TaiESM




**Figure List**

**Figure 1.** Peak phase of diurnal rainfall cycle over three major tropical regions: Central Africa,
Southeast Asia, and Amazonia in (a) TRMM3B42 (2001–2011), (b) CESM1.2.2 (1979–2005), and (c)
TaiESM (1979–2005).

**Figure 2.** Time-longitude Hovmöller diagrams for diurnal rainfall cycle over the SGP observed by
TRMM3B42 dataset (2001–2011, upper panel), and simulated by CESM1.2.2 (central panel) and
TaiESM (lower panel), with the elevation of topography on the top.

**Figure 3.** Theoretical calculations of cloud fraction as a function of RH for water vapor and
condensates: (a) CAM5 macrophysics scheme, (b) GTS macrophysics with uniform PDF, and (c)
GRS macrophysics with triangular PDF.

**Figure 4.** A 500-year time series of annual mean climatological quantities in TaiESM piControl
simulation (from top to bottom): SAT at 2-m height, SST, net flux at the TOA (FNT), net flux at the
surface (FNS), SSS, volume-averaged ocean temperature, volume-averaged ocean salinity, and NH
and SH sea ice areas. The horizontal lines in FNT and FNS indicate the zero value.

**Figure 5.** Historical global mean SAT anomalies relative to the period of 1951–1980 from TaiESM
historical simulation (red) and observational datasets of BEST (blue) and GISTEMP (black).

**Figure 6.** Vertically integrated cloud fractions for (a) total cloud, (b) high cloud, and (c) low cloud in
the 1979–2005 TaiESM historical run (top panels), observations (MODIS for total cloud and
CloudSat–CALIOP for high and low cloud, central panels) and biases (bottom panels).




**Figure 7.** Cloud forcing for (a) shortwave and (b) longwave in the 1979–2005 TaiESM historical run
(top panels), observations (central panels, CERES–EBAF), and biases (bottom panels).

**Figure 8.** (a) SST and (b) SAT in the 1979–2005 TaiESM historical run (top panels), observations
(HadISST for SST and BEST for SAT, central panels), and biases (bottom panels).

**Figure 9.** Precipitation in the 1979–2005 TaiESM historical run (top panels), observations (GPCP,
central panels), and biases (bottom panels).

**Figure 10.** Annual mean sea ice concentration in the 1979–2005 TaiESM historical run for both NH
and SH. The solid black lines indicate the 15% sea ice concentration from the observation (NSIDC–
CDR, 1979–2005).

**Figure 11.** Time series of annual mean total sea ice area for both NH and SH from TaiESM
historical run and observation.

**Figure 12.** The space-time RMSEs of upward longwave radiation at TOA in total sky and clear sky
(RLUT and RLUTCS), upward shortwave radiation at TOA in total sky and clear sky (RSUT and
RSUTCS), precipitation (PR), surface air temperature (TAS), geopotential height (ZG), meridional
wind (VA), zonal wind (UA), and air temperature (TA) from TaiESM, CMIP5 models, and CMIP5
MME. The values of shading represent the magnitude of normalized error with respect to the median
CMIP5 error. For example, a value of -0.2 indicates that the RMSE of a model is 20% smaller than
the median error.



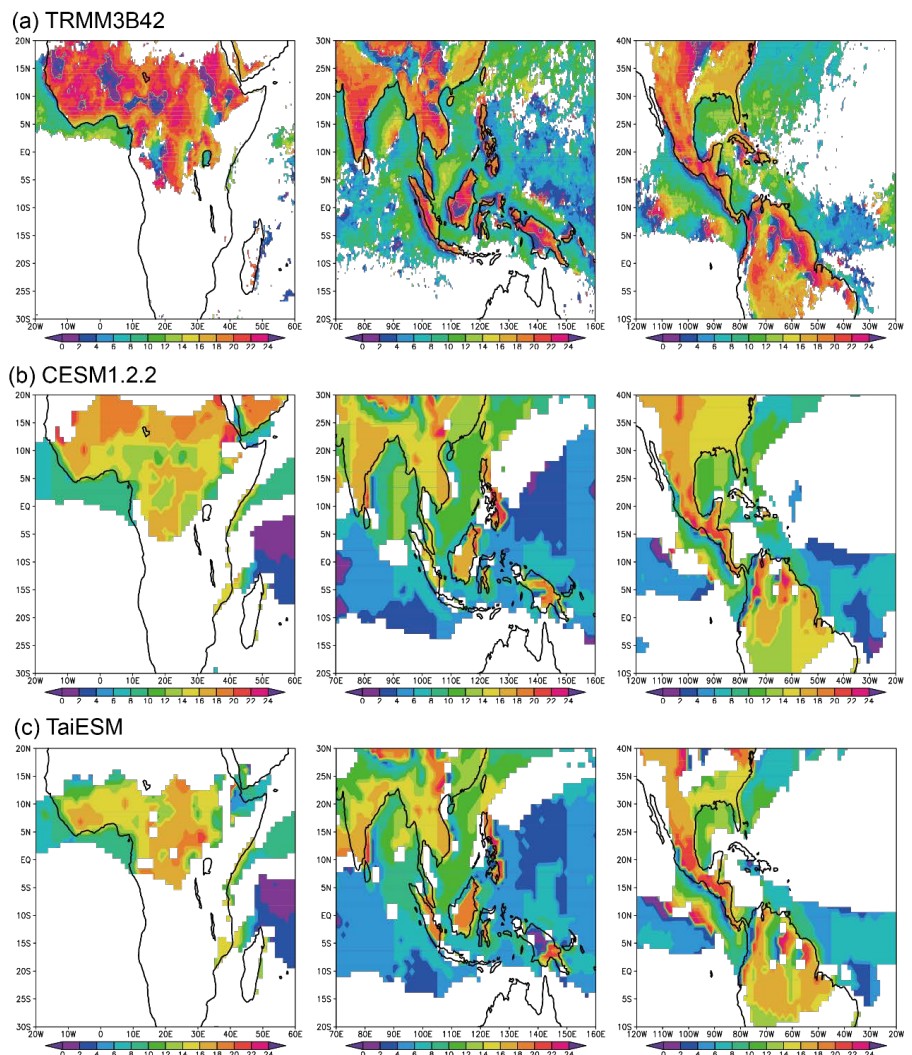

**Figure 1.** Peak phase of diurnal rainfall cycle over three major tropical regions: central Africa, Southeast Asia, and Amazonia in (a) TRMM3B42 (2001–2011), (b) CESM1.2.2 (1979–2005), and (c) TaiESM (1979–2005).



## Southern Great Plains (35-40N,90-110W)

**Figure 2.** Time-longitude Hovmöller diagrams for diurnal rainfall cycle over the SGP observed by TRMM3B42 dataset (2001–2011, upper panel), and simulated by CESM1.2.2 (central panel) and TaiESM (lower panel), with the elevation of topography on the top.



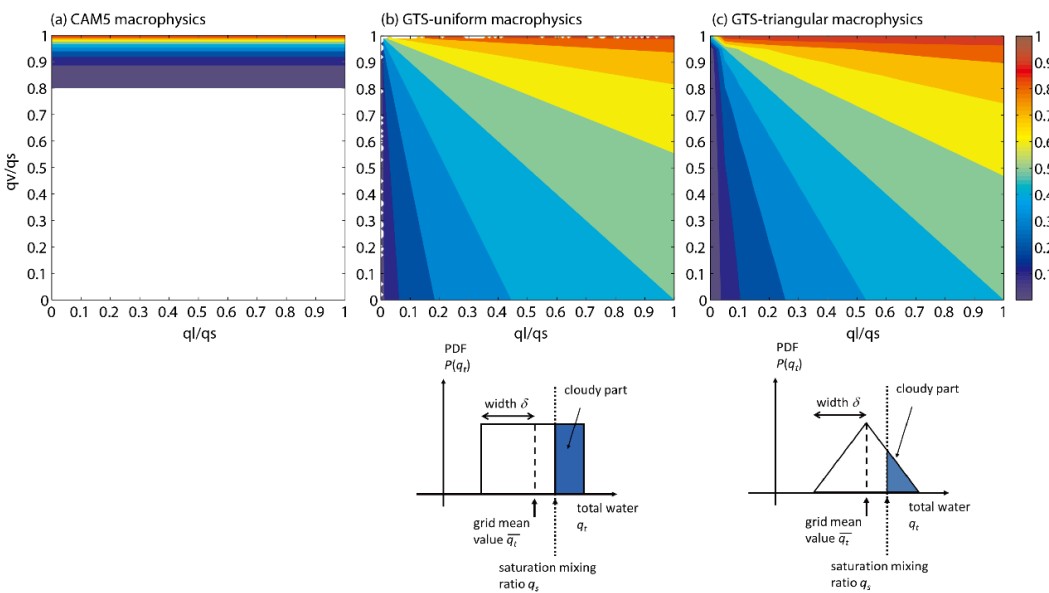

**Figure 3.** Theoretical calculations of cloud fraction as a function of RH for water vapor and condensates: (a) CAM5 macrophysics scheme, (b) GTS macrophysics with uniform PDF, and (c) GRS macrophysics with triangular PDF.




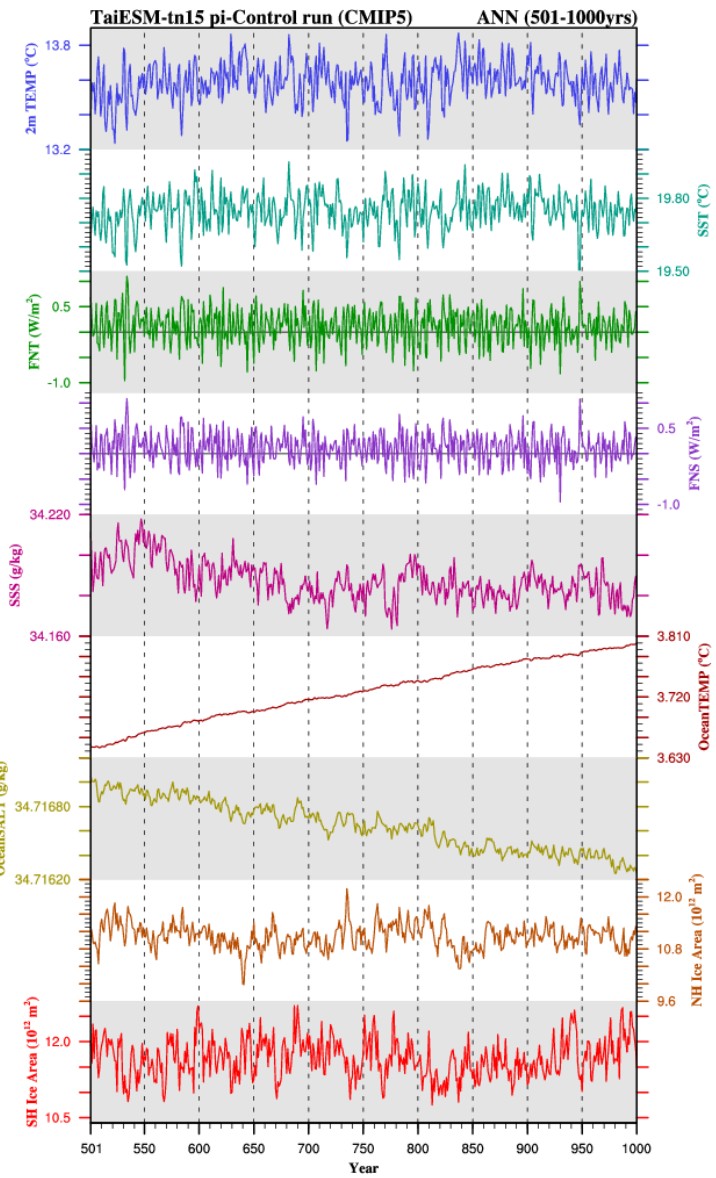

**Figure 4.** A 500-year time series of annual mean climatological quantities in TaiESM piControl
simulation (from top to bottom): SAT at 2-m height, SST, net flux at the TOA (FNT), net flux at the
surface (FNS), SSS, volume-averaged ocean temperature, volume-averaged ocean salinity, and NH
and SH sea ice area. The horizontal lines in FNT and FNS indicate the zero value.





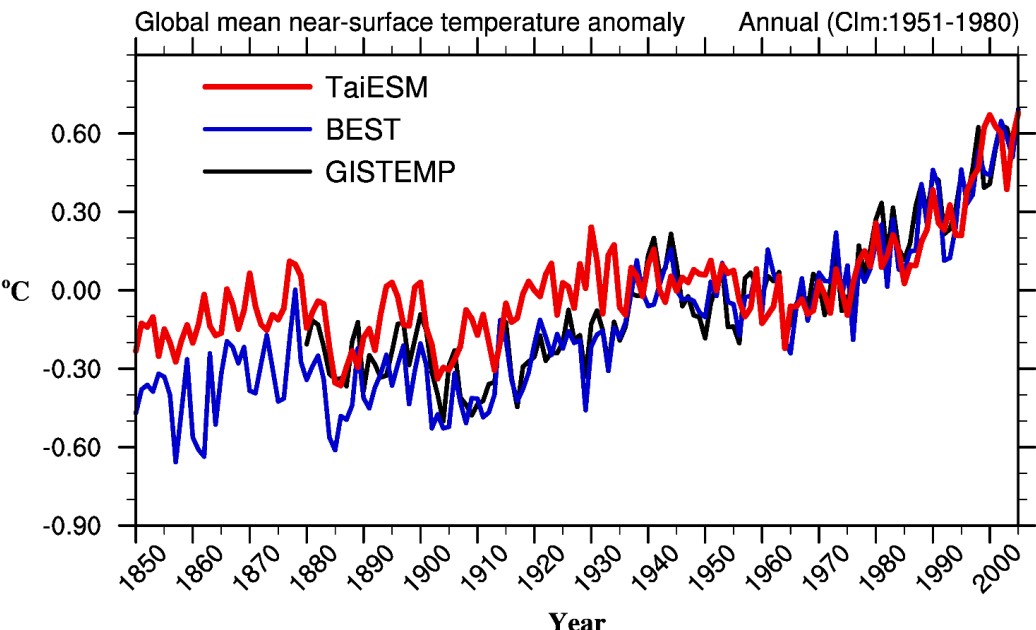


**Figure 5.** Historical global mean SAT anomalies relative to the period of 1951–1980 from TaiESM

historical simulation (red) and observational datasets of BEST (blue) and GISTEMP (black).


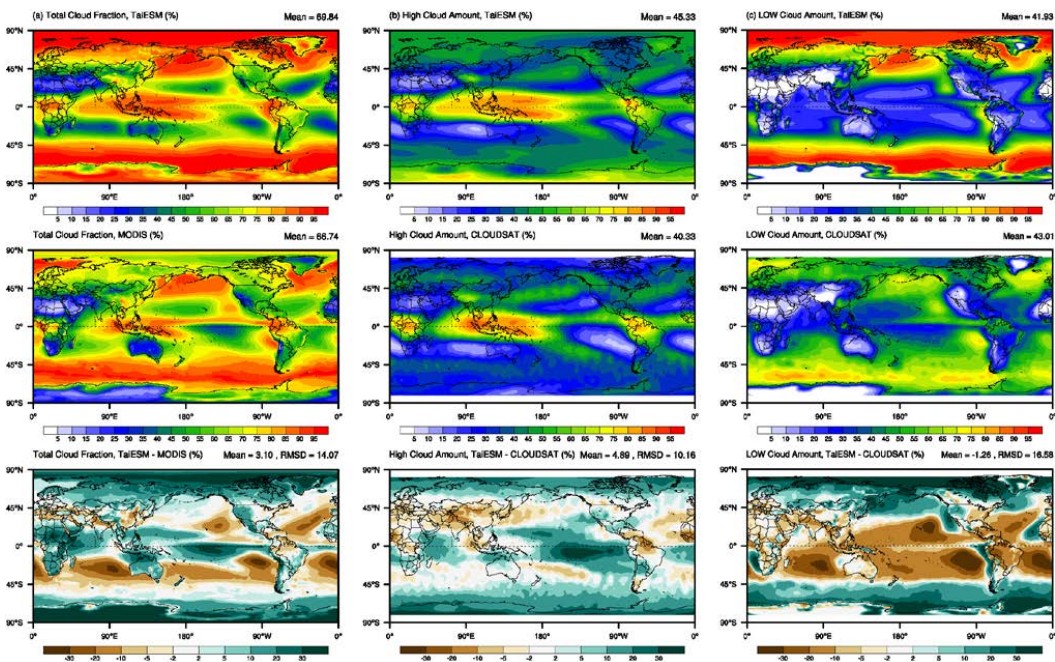

**Figure 6.** Vertically integrated cloud fractions for (a) total cloud, (b) high cloud, and (c) low cloud in the 1979–2005 TaiESM historical run (top panels), observations (MODIS for total cloud and CloudSat–CALIOP for high and low cloud, central panels) and biases (bottom panels).

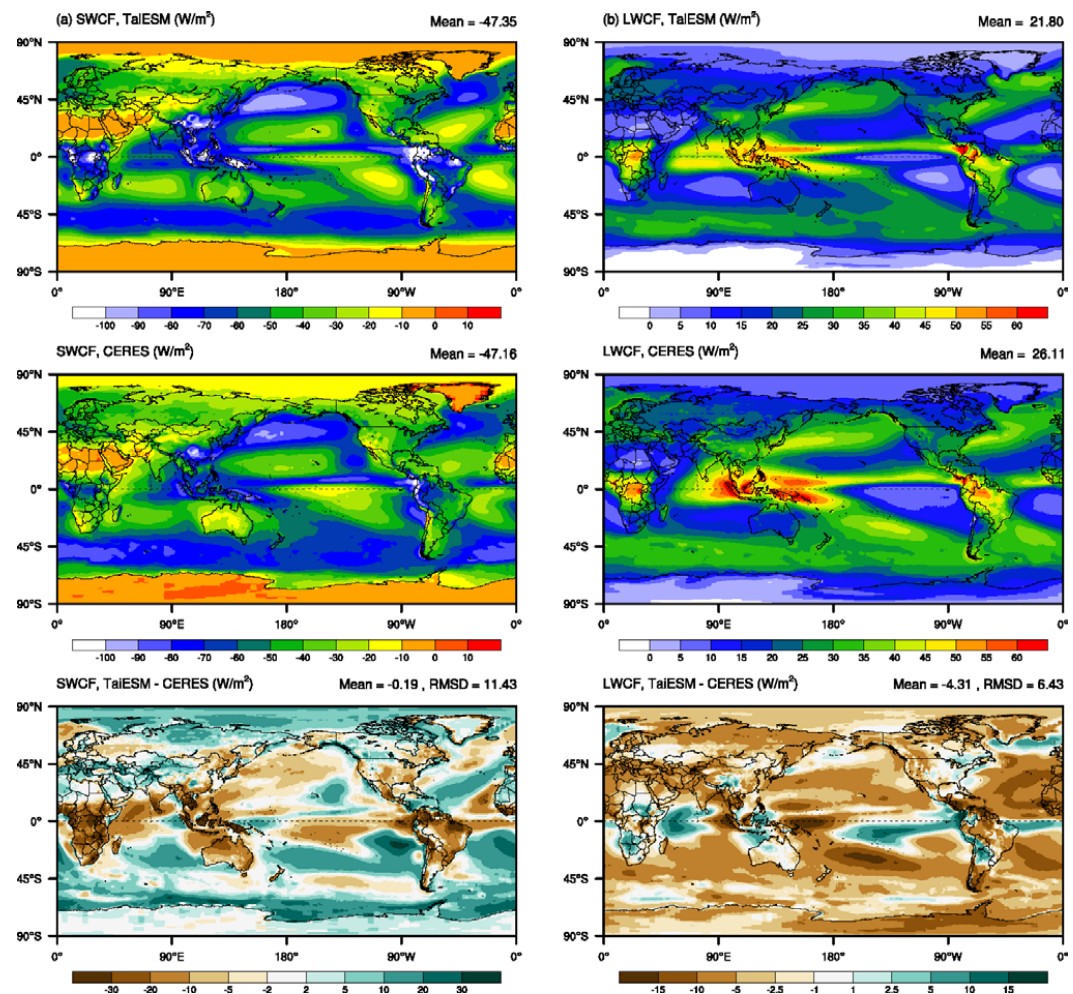


**Figure 7.** Cloud forcing for (a) shortwave and (b) longwave in the 1979–2005 TaiESM historical run


(top panels), observations (central panels, CERES–EBAF), and biases (bottom panels).



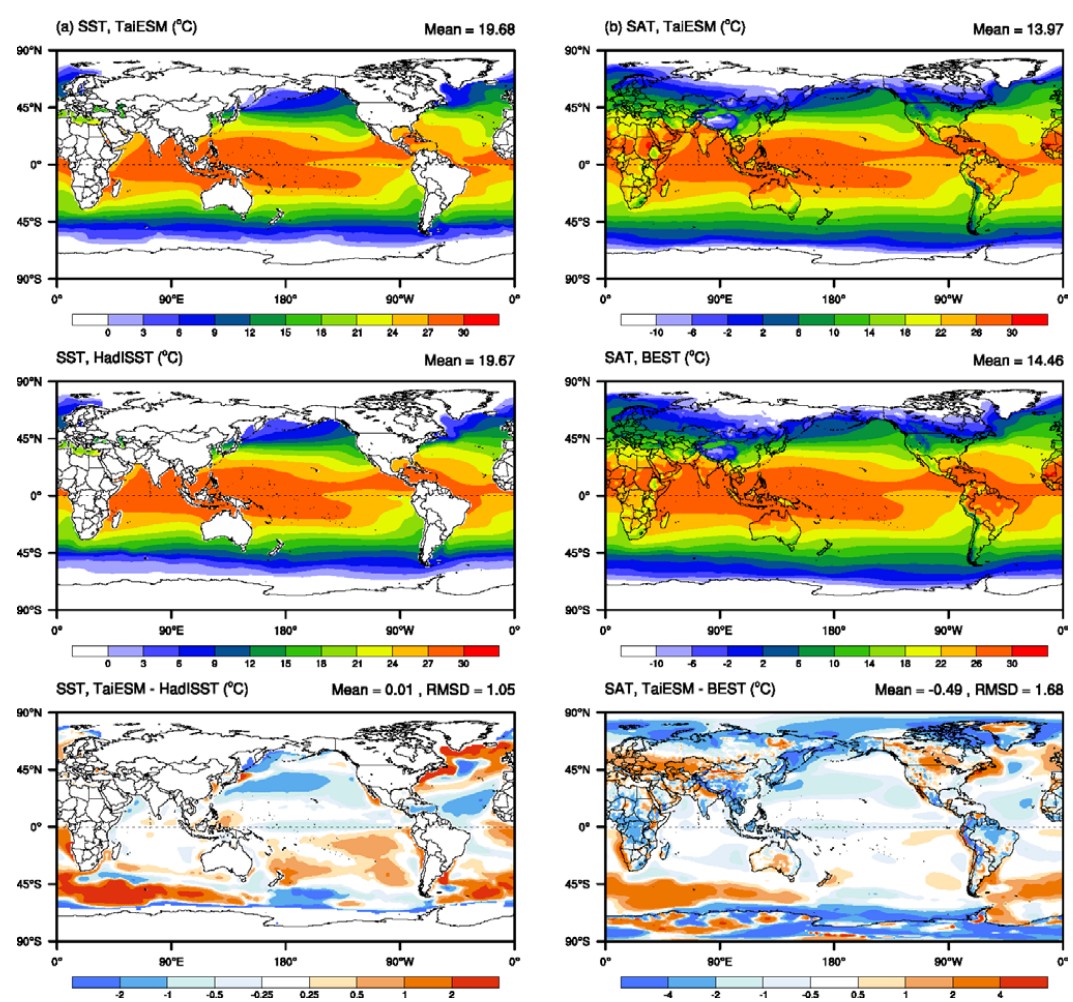

**Figure 8.** (a) SST and (b) SAT in the 1979–2005 TaiESM historical run (top panels), observations

(HadISST for SST and BEST for SAT, central panels), and biases (bottom panels).





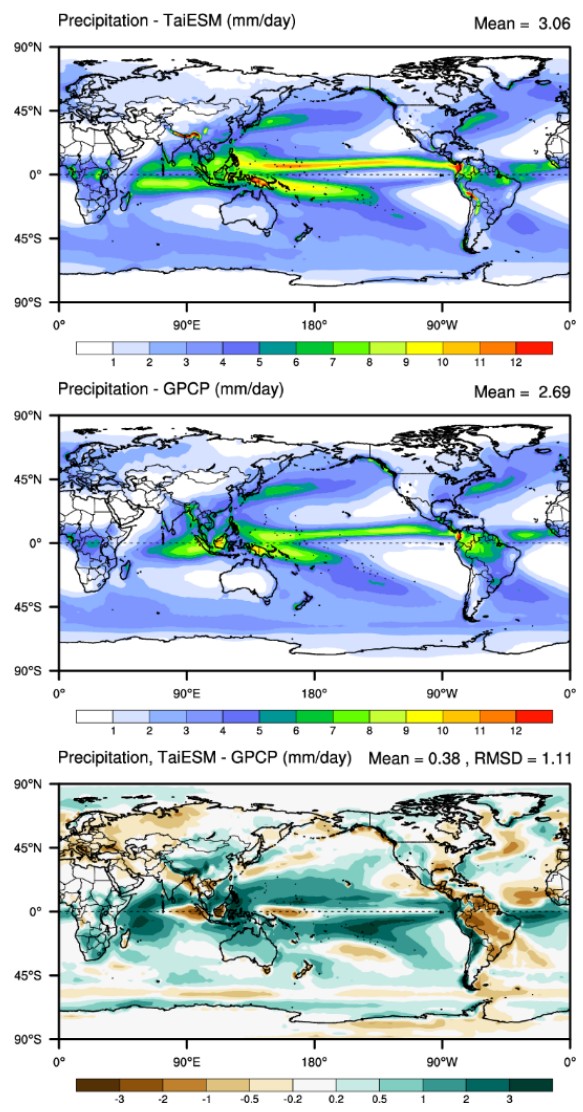

**Figure 9.** Precipitation in the 1979–2005 TaiESM historical run (top panels), observations (GPCP,

central panels), and biases (bottom panels).

760



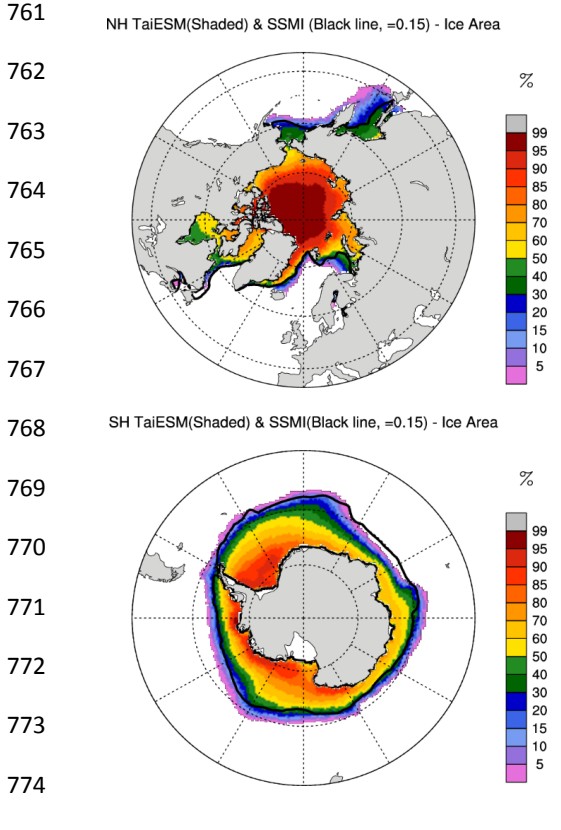

**Figure 10.** Annual mean sea ice concentration in the 1979–2005 TaiESM historical run for both NH and SH. The solid black lines indicate the 15% sea ice concentration from the observation (NSIDC–CDR, 1979–2005).





781

**Figure 11.** Time series of annual mean total sea ice area for both NH and SH from TaiESM

historical run and observation.

784



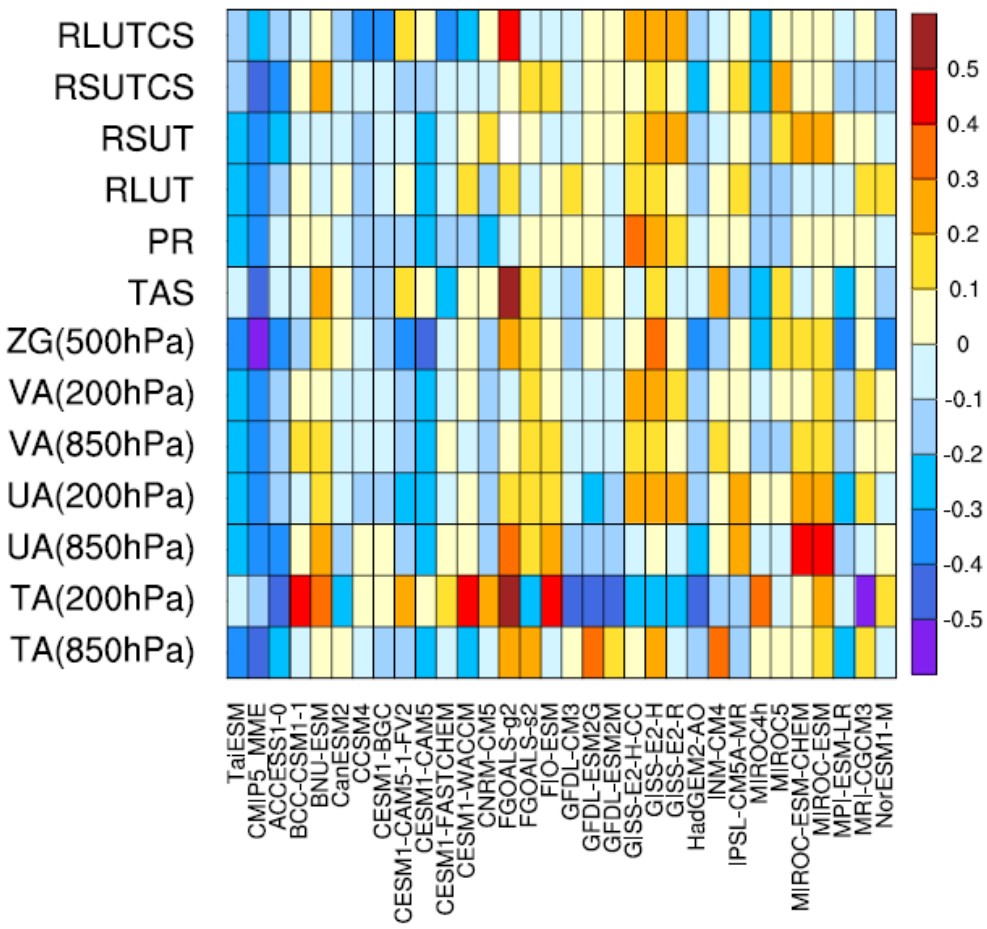

785

**Figure 12.** The space-time RMSEs of upward longwave radiation at TOA in total sky and clear sky

(RLUT and RLUTCS), upward shortwave radiation at TOA in total sky and clear sky (RSUT and

RSUTCS), precipitation (PR), surface air temperature (TAS), geopotential height (ZG), meridional

wind (VA), zonal wind (UA), and air temperature (TA) from TaiESM, CMIP5 models, and CMIP5

MME. The values of shading represent the magnitude of normalized error with respect to the median

CMIP5 error. For example, a value of -0.2 indicates that the RMSE of a model is 20% smaller than

the median error.