# Peer review of "Taiwan Earth System Model Version 1: Description and Evaluation of Mean State"

_Geoscientific Model Development, 2019_

## Referee Comment (RC1) · Anonymous Referee #1 · 25 Feb 2020

This is a description paper of the new Taiwan Earth System Model (TaiESM) Version 1, which is developed based on CESM1.2.2. Updated physics from CESM are described and the basic features in mean climates of TaiESM are shown. Model biases are also compared with CMIP5 models. As such, this paper should appear in GMD. Some minor addition and corrections would improve the quality of the paper.

(1) Although the paper only deals with mean state, information of simulated ENSO would be desirable because ENSO is the largest important of tropical variability of the atmosphere-ocean system. This has been included in most of description papers.

(2) line 118-120: Mention that large bias still exists compared to TRMM3B42. TRMM shows 20-22 LT peak, while model shows 16-18 LT peak over Africa, India and Indochina peninsula.

(3) Line 135: What is "solesoid"? Solenoid?

(4) Line 221: Where is the model top. Add "model top at x.xx hPa"

(5) Line 279, Table 1: Add estimated observed values.

(6) Line 327-333: Figure 6 shows that there are positive bias near the coast and negative bias off the coast in low cloud fraction. Is there any explanation for this?

(7) Line 413-415: This point should be described in Section 2 Model description. Similarly, is river runoff treated as input in the ocean model?

(8) Line 418-434: Also mention whether TaiESM is better the CESM or comparable or other.

(9) Line 331: Figure 6b should be Figure 6c.

(10) Line 333: Figure 6c should be Figure 6b.

(11) There are two Wang et al. (2015). Distinguish the two in the main text and references.

(12) Some references use "and co-authors", while others write down all authors. Follow the journal requirement.

(13) A long list of incomplete referencing. Followings are in the main text, but not in the reference list. IPCC 5th Assessment Report 2013 Wang et al. 2018 > 2019? Shiu et al. 2018 Pan and Wu 1995 Han and Pan 2011 Tu et al. 2005: may be Tu and Tsuang 2005 Park and Bretherton 2009 Bretherton and Park 2009 Morrison and Gettelmen 2008 Zhang and McFarlane 1999 > 1995? Shamrock et al. 2008 > 2005? Gu et al. 2012 Liou et al. 2013 Hunke and Lipscomb 2008 Smith et al. 2010 Kay and Gettelman 2009 Gent et al. 2011 Following is in the reference list, but not in the main text. Guichard et al. 2004
* * *
[Figure]

2020.

---

## Referee Comment (RC2) · Anonymous Referee #2 · 9 May 2020

The paper describes and evaluates the first version of the Taiwan Earth System Model. The model is derived from NCAR's Community Earth System Model version 1.2, with specific parts being replaced and modified in order to optimize its performance for the East and Southeast Asia region.

The main value of TaiESM is thus in regional application and to serve as an ESM infrastructure that facilitates integrating national climate research efforts. In addition, there can be a broader interest in using TaiESM and its output, as its closeness to the host system provides an opportunity for studying how specific changes in model representations affect regional and global biases. In the long term, innovations developed in TaiESM may also feed back to the host model.

The description part of the paper focuses on the modifications and new innovations,

which are motivated from regional precipitation deficiencies of the CESM host system. The evaluation part provides a non-exhaustive but sufficient general evaluation (model stability, global climate sensitivity, spatial surface biases etc.) and some more specific evaluation such as the diurnal cycle of precipitation. The paper offers a measured balance between highlighting the strengths of TaiESM and documenting its weaknesses.

The paper is well written and I expected it to become a key reference for further model development of TaiESM as well as inform any studies using the model or its output. It is thus well within the scope of the journal and I recommend its publication. I have only some very minor specific comments that the authors may want to consider:

L73-75 language suggestion: "account" -> "accounts", "for application" -> "and designed for application"

L107-108 "Wang et al. (2015) reported significant improvements": One could state here which model system (or at least which type of model system) Wang et al. used.

L122-124 "where propagating convective organizations emitting from the coastline or topographical regions (Kikuchi and Wang, 2010), demonstrated as the gradual phase change in Figure 1": unclear formulation

L155 "grid" -> "grid box"

L156 "by two PDFs" -> "by the two PDFs"

L157 "The triangular PDF provide" -> "The triangular PDF provides"

L164 "adapted" or "adopted" ?

L223-224 "several microphysical properties of clouds are modified to minimize radiation imbalance": Was the TOA imbalance positive or negative before re-tuning? It could be interesting for readers to know which properties exactly were re-tuned and in which direction these were tuned.

L249 "the less imbalance" -> "the comparatively less imbalance"

L251 "0.0088 K century-1 in 500 years, which is significant": Maybe change to "is statistically significant"? The trend is very small and seems insignificant for most practical applications.

L274-275 "In addition, the long-term mean of evaporation minus precipitation (E − P) is −1.16 mm day-1, and it may also contribute to the freshening of the ocean.": I would think this is the main reason of the ocean freshening. An E-P imbalance of -1 mm/day corresponds to quite a few meters per century and it could be an idea to caution the reader (either here or in the summary section) to account for this drift when using TaiESM output for sea level studies.

L290-291 "greater contribution to addition": unclear formulation

L293-294 "The relation between ... must be due to": Should this be "The different relation between... must be due to" ?

L297 "with the" -> "against" ?

L340-341 " SWCF is not as strong as that in the observational data. It indicates that polar cloud in TaiESM is too thin optically": Could sea ice/snow albedo bias potentially also contribute to weaker SWCF?

L413 "no land–sea model": What does that mean? No dynamic land ice model and/or ice shelf model?

L434 "." missing at end of sentence

L458 "almost similar": Do you mean "mostly similar"?

Figure 1: The differences between CESM1.2.2 and TaiESM are hard to discern by eye when comparing (b) and (c). It therefore could be an idea to add another row of panels that shows the TaiESM - CESM1.2.2 differences. Also, given that CESM1.2.2 and TaiESM use the same horizontal grid it is somewhat surprising that the regions with missing value in (b) don't match those in (c).

---

## Author Comment (AC1) · 22 Jun 2020

This is a description paper of the new Taiwan Earth System Model (TaiESM) Version 1, which is developed based on CESM1.2.2. Updated physics from CESM are described and the basic features in mean climates of TaiESM are shown. Model biases are also compared with CMIP5 models. As such, this paper should appear in GMD. Some minor addition and corrections would improve the quality of the paper.

**Response:** We thank the reviewer for the positive comments. Below please see our point-by-point responses. The line numbers correspond to the change-tracked manuscript.

(1) Although the paper only deals with mean state, information of simulated ENSO would be desirable because ENSO is the largest important of tropical variability of the atmosphere-ocean system. This has been included in most of description papers.

**Response:** Following the reviewer's suggestion, a subsection about the performance of ENSO simulation in TaiESM is added in Lines 452-469 with two figures.

(2) line 118-120: Mention that large bias still exists compared to TRMM3B42. TRMM shows 20-22 LT peak, while model shows 16-18 LT peak over Africa, India and Indochina peninsula.

**Response:** Following the reviewer's suggestion, the occurrence times of diurnal rainfall peaks in the observations are added. (Lines 122-123)

(3) Line 135: What is "solesoid"? Solenoid?

**Response:** Yes, it is corrected. (Line 142)

(4) Line 221: Where is the model top. Add "model top at x.xx hPa"

**Response:** We have added "model top at 2 hPa" in the text. (Line 254)

(5) Line 279, Table 1: Add estimated observed values.

**Response:** The observation values of SST and SAT are added as a note in Table 1.

(6) Line 327-333: Figure 6 shows that there are positive bias near the coast and negative bias off the coast in low cloud fraction. Is there any explanation for this?

**Response:** We believe it is caused by the coarse resolution (~2°) of the observation data.

(7) Line 413-415: This point should be described in Section 2 Model description. Similarly, is river runoff treated as input in the ocean model?

**Response:** Following the reviewer's suggestion, we have added descriptions of River Transport Model and the lack of a land ice model. (Lines 207-211)

(8) Line 418-434: Also mention whether TaiESM is better the CESM or comparable or other.

**Response:** Following the reviewer's suggestion, we have added two sentences about the comparison with CESM: "The performance of TaiESM is comparable to that of CESM1-CAM5, and they have similar strengths and weaknesses. Note that three variables below average in CESM1-CAM5 are all improved in TaiESM." (Lines 487-489)

(9) Line 331: Figure 6b should be Figure 6c.

**Response:** Corrected. (Line 363)

(10) Line 333: Figure 6c should be Figure 6b.

**Response:** Corrected. (Line 365)

(11) There are two Wang et al. (2015). Distinguish the two in the main text and references.

**Response:** Done.

(12) Some references use "and co-authors", while others write down all authors. Follow the journal requirement.

**Response:** Done.

(13) A long list of incomplete referencing. Followings are in the main text, but not in the reference list. IPCC 5th Assessment Report 2013 Wang et al. 2018 > 2019? Shiu et al. 2018 Pan and Wu 1995 Han and Pan 2011 Tu et al. 2005: may be Tu and Tsuang 2005 Park and Bretherton 2009 Bretherton and Park 2009 Morrison and Gettelmen 2008 Zhang and McFarlane 1999 > 1995? Shamrock et al. 2008 > 2005? Gu et al. 2012 Liou et al. 2013 Hunke and Lipscomb 2008 Smith et al. 2010 Kay and Gettelman 2009 Gent et al. 2011 Following is in the reference list, but not in the main text. Guichard et al. 2004

**Response:** Thank you for this detailed list. We have made all necessary corrections accordingly.

---

## Author Comment (AC3) · 22 Jun 2020

The paper describes and evaluates the first version of the Taiwan Earth System Model. The model is derived from NCAR's Community Earth System Model version 1.2, with specific parts being replaced and modified in order to optimize its performance for the East and Southeast Asia region.

The main value of TaiESM is thus in regional application and to serve as an ESM infrastructure that facilitates integrating national climate research efforts. In addition, there can be a broader interest in using TaiESM and its output, as its closeness to the host system provides an opportunity for studying how specific changes in model representations affect regional and global biases. In the long term, innovations developed in TaiESM may also feed back to the host model.

The description part of the paper focuses on the modifications and new innovations, which are motivated from regional precipitation deficiencies of the CESM host system. The evaluation part provides a non-exhaustive but sufficient general evaluation (model stability, global climate sensitivity, spatial surface biases etc.) and some more specific evaluation such as the diurnal cycle of precipitation. The paper offers a measured balance between highlighting the strengths of TaiESM and documenting its weaknesses.

The paper is well written and I expected it to become a key reference for further model development of TaiESM as well as inform any studies using the model or its output. It is thus well within the scope of the journal and I recommend its publication. I have only some very minor specific comments that the authors may want to consider:

**Response:** We thank the reviewer for the comprehensive summary and encouraging comments. Below please see our point-by-point responses. The line numbers correspond to the change-tracked manuscript.

L73-75 language suggestion: "account" -> "accounts", "for application" -> "and designed for application"

**Response:** Corrected. (Lines 73-35)

L107-108 "Wang et al. (2015) reported significant improvements": One could state here which model system (or at least which type of model system) Wang et al. used.

**Response:** CESM1.0.3 with CAM5.1 was used in their paper. This info is added to the text (Line 107).

L122-124 "where propagating convective organizations emitting from the coastline or topographical regions (Kikuchi and Wang, 2010), demonstrated as the gradual phase change in Figure 1": unclear formulation

**Response:** We have rewritten this and next paragraphs to make it clearer. (Lines 125-132)

L155 "grid" -> "grid box"
L156 "by two PDFs" -> "by the two PDFs"
L157 "The triangular PDF provide" -> "The triangular PDF provides"

**Response:** These errors are corrected. (Lines 163-166)

L164 "adapted" or "adopted" ?

**Response:** It is "adopted". (Line 172)

L223-224 "several microphysical properties of clouds are modified to minimize radiation imbalance": Was the TOA imbalance positive or negative before re-tuning? It could be interesting for readers to know which properties exactly were re-tuned and in which direction these were tuned.

**Response:** Following the reviewer's suggestion, we have added Section 2.4 to discuss more details about tuning TaiESM. (Lines 231-250)

L249 "the less imbalance" -> "the comparatively less imbalance"

**Response:** Done. (Line 281)

L251 "0.0088 K century-1 in 500 years, which is significant": Maybe change to "is statistically significant"? The trend is very small and seems insignificant for most practical applications.

**Response:** Done. (Line 283)

L274-275 "In addition, the long-term mean of evaporation minus precipitation (E - P) is -1.16 mm day-1, and it may also contribute to the freshening of the ocean.": I would think this is the main reason of the ocean freshening. An E-P imbalance of -1 mm/day corresponds to quite a few meters per century and it could be an idea to caution the reader (either here or in the summary section) to account for this drift when using TaiESM output for sea level studies.

**Response:** We thank the reviewer to point out that the value of (E-P) is too large. We deleted this sentence because there is large uncertainty in estimating (E-P). TaiESM does not directly output the amount of evaporation, and we calculated it using the surface latent heat flux. However, the magnitude of uncertainty in this calculation is probably larger than the difference between evaporation and precipitation. Therefore, we decide not to discuss the issue related to the change in global salinity here.

L290-291 "greater contribution to addition": unclear formulation
**Response:** "addition" is revised to "additional". (Line 322)

L293-294 "The relation between ... must be due to": Should this be "The different relation between... must be due to" ?
**Response:** Done. (Line 324)

L297 "with the" -> "against" ?
**Response:** Done. (Line 329)

L340-341 " SWCF is not as strong as that in the observational data. It indicates that polar cloud in TaiESM is too thin optically": Could sea ice/snow albedo bias potentially also contribute to weaker SWCF?
**Response:** We examined the bias of surface albedo over the Arctic Ocean and found that while the sea ice extent in TaiESM is lower than observation, albedo in TaiESM is larger. Therefore, it does contribute to the smaller SWCF over the Arctic Ocean in TaiESM. We have modified this sentence in Lines 372-375.

L413 "no land–sea model": What does that mean? No dynamic land ice model and/or ice shelf model?
**Response:** "land-sea model" should be "land ice model" and corrected. (Line 447)

L434 "." missing at end of sentence
**Response:** This sentence is removed.

L458 "almost similar": Do you mean "mostly similar"?
**Response:** Corrected. (Line 513)

Figure 1: The differences between CESM1.2.2 and TaiESM are hard to discern by eye when comparing (b) and (c). It therefore could be an idea to add another row of

panels that shows the TaiESM - CESM1.2.2 differences. Also, given that CESM1.2.2 and TaiESM use the same horizontal grid it is somewhat surprising that the regions with missing value in (b) don't match those in (c).

**Response:** We found that plotting the differences in precipitation peaks makes the figure quite noisy. Therefore, we have added several boxes in the figure to highlight the area with remarkable differences.

For the missing value, we masked out the areas with the amplitude of diurnal precipitation smaller than 0.8 mm day$^{-1}$ in the original figure. To make the figure clearer, we lower the threshold to 0.5 mm day$^{-1}$. (Line 117)